# Topographic Distribution of miRNAs (miR-30a, miR-223, miR-let-7a, miR-let-7f, miR-451, and miR-486) in the Plasma Extracellular Vesicles

**DOI:** 10.3390/ncrna10010015

**Published:** 2024-02-15

**Authors:** Tatiana Petrova, Olga Kalinina, Arthur Aquino, Evgeniy Grigoryev, Natallia V. Dubashynskaya, Kseniya Zubkova, Anna Kostareva, Alexey Golovkin

**Affiliations:** 1Almazov National Medical Research Centre, 197341 St. Petersburg, Russia; tatianakhod@gmail.com (T.P.); olgakalinina@mail.ru (O.K.); akino97@bk.ru (A.A.); zubkowa.ksenija@gmail.com (K.Z.); akostareva@hotmail.com (A.K.); 2St. Petersburg State University, Research Park, 199034 St. Petersburg, Russia; grigoryev_egeniy@mail.ru; 3Institute of Macromolecular Compounds of the Russian Academy of Sciences, 199004 St. Petersburg, Russia; dubashinskaya@gmail.com

**Keywords:** extracellular vesicles, microvesicles, miRNA, miRNAs plasma levels, miRNA distribution, extracellular vesicles’ cargo

## Abstract

There are many articles on the quantitative analysis of miRNAs contained in a population of EVs of different sizes under various physiological and pathological conditions. For such analysis, it is important to correctly quantify the miRNA contents of EVs. It should be considered that quantification is skewed depending on the isolation protocol, and different miRNAs are degraded by nucleases with different efficiencies. In addition, it is important to consider the contribution of miRNAs coprecipitating with the EVs population, because the amount of miRNAs in the EVs population under study is skewed without appropriate enzymatic treatment. By studying a population of EVs from the blood plasma of healthy donors, we found that the absolute amount of miRNA inside the vesicles is commensurate with the amount of the same type of miRNA adhered to the outside of the EVs. The inside/outside ratio ranged from 1.02 to 2.64 for different investigated miRNAs. According to our results, we propose the hypothesis that high occupancy of miRNAs on the outer surface of EVs influence on the transporting RNA repertoire no less than the inner cargo received from the host cell.

## 1. Introduction

Extracellular vesicles (EVs) are universal naturally released bilipid membrane structures of different sizes that form in both eukaryotes and prokaryotes [1,2,3]. According to their size and biogenesis pathways, eukaryotic EVs can be divided into apoptotic bodies, microvesicles, or ectosomes and exosomes [4]. Apoptotic bodies are formed as a result of programmed cell death and are 800 to 5000 nm in size. Microvesicles or ectosomes are formed by budding from the cell membrane and range in size from 150 to 1000 nm. Exosomes are formed from multivesicular bodies that fuse with the cell membrane, releasing their contents to the outside, and they range in size from 30 to 150 nm [5,6]. However, the International Society for Extracellular Vesicles proposes classifying EVs by size, biochemical composition, and cell of origin, since the identification of the precise biogenesis of different EV types is complicated [7].

It is known that EVs transport various biologically active molecules, such as proteins, lipids, nucleic acids, and metabolites, from host cells to recipient cells, carrying out a powerful intercellular communication [3,6,8]. The delivery of biologically active molecules can be carried out both to neighboring cells and to those distant from the donor cell [9]. This intercellular communication can have a wide range of effects in the body, from the regulation of physiological processes to the maintenance of homeostasis [4]. Beyond that, there are numerous papers describing the involvement of EVs in various pathological conditions [10,11,12,13].

In addition to other molecules, EVs contain different types of RNA, such as mRNA, rRNA, lncRNA, miRNA, piwi-RNA, and others. The RNA profile varies based on the vesicle type and the characteristics of the EV’s parental cells [14,15,16,17,18,19,20]. RNAs carried by EVs can affect gene expression in recipient cells, for example, by miRNAs [20]. Moreover, some miRNAs have demonstrated a pathological role in EVs. For instance, miRNA 105, secreted in breast cancer cell exosomes, disrupts epithelial monolayer tight junctions, promoting metastasis [21]. EVs released by lung-cancer-activated mast cells contain significantly increased amounts of protumorigenic miRNA 100 and miRNA 125b, which regulate p53 signaling, cancer pathways and pathways related to apoptosis and the cell cycle [22].

Mature miRNAs are short, conserved, noncoding RNAs 18–22 nucleotides long that regulate gene expression by inhibiting complementary mRNAs [23,24]. They have been shown to regulate essential biological processes in the cell, including proliferation, differentiation, cell-cycle regulation, apoptosis, and many others [25,26,27]. Extracellular miRNAs can be present in plasma inside the EVs or as a part of free circulating complexes with Ago2 proteins and other members of the Ago family, or with high-density lipoproteins [28,29,30,31]. All miRNAs found in human plasma are very stable because they are protected from blood nucleases by EVs or protein complexes [11,32,33].

MiRNAs transported in EVs or in complex with high-density lipoproteins affect recipient cells by regulating protein expression in them [31,34,35]. It is known that the cell can regulate the concentration of miRNA in the vesicles that it releases, and several mechanisms for selective sorting have already been described, including ESCRT, tetraspanins, and lipid-dependent mechanisms [8,35,36,37,38].

The study of the molecular composition of the contents of EVs is complicated by the lack of standardized methods for sample preparation and analysis so that results from different research groups can be correctly compared and new knowledge can be synthesized. In particular, we were interested in the contribution of the amount of analyzed miRNA that coprecipitates with the EV population during differential centrifugation. In the present work, using RT qPCR, we measured the absolute amounts of some miRNAs inside EVs and adhered externally to EVs or coprecipitated with EVs. The following miRNAs were investigated: hsa-miR-let-7a, hsa-miR-let-7f, hsa-miR-30a, hsa-miR-486, hsa-miR-223, and hsa-miR-451. MiRNA-223 was chosen as one of the most abundant miRNAs expressed in myeloid cells [39]. MiR-451 participates in erythroid maturation and differentiation [40]. MiRNA-486 is a muscle-associated miRNA that is highly abundant in human plasma and enriched in small extracellular vesicles [41]. MiRNA 30a participates in differentiation and development [42]. MiRNA let-7a and miRNA let-7f belong to the conservative let-7 family [43,44].

## 2. Results

### 2.1. Characteristics of MV Concentrates of Healthy Volunteers’ Blood Plasma

In our work, we used differential centrifugation as one of the most common methods for isolating EVs from various biological fluids. In order to purify the EV fraction of human plasma from platelet debris, we filtered the plasma through an 800 nm filter after differential centrifugation (Figure 1). However, our samples contained impurities of highly abundant plasma proteins, lipoproteins, and immune complexes that were precipitated simultaneously with the EVs [45,46,47].

### 2.2. Enzymatic Treatments

In the present study, we aimed to determine the absolute amounts of miRNAs (has-miR-let-7a, has-miR-let-7f, has-miR-30a, hsa-miR-486, has-miR-223, has-miR-451) from human blood plasma both loaded in the investigated EVs and adhered to the EVs externally. For this purpose, we performed a series of combined and separate treatments of EV population with detergent Triton X 100, proteinase K, and RNAse A. Since the interaction of miRNA with Ago2 protects the complex from the action of RNAse A, the pretreatment with proteinase K was necessary [48]. Three samples of MVs were obtained from every donor, each concentrated from 1 mL of pretreated plasma (see Methods). The first sample from each donor was consequently treated with Triton X 100, proteinase K, and RNAse A, the second one with proteinase K and RNAse A, and the third one was not treated (Figure 1). During the first treatment, we expected sequential destruction of the bilipid membrane (Triton X 100), proteins that protected miRNAs from the nucleases (proteinase K), and the miRNAs themselves (RNAse A). During the second treatment, we expected destruction of proteins and miRNAs that were not protected by the bilipid membrane of the EVs. The third samples were used as controls containing both miRNAs inside the EVs and adhered externally.

### 2.3. Western Blot

Depending on the type of specimen under study, the combinations of the following protein markers of EVs are often used: tetraspanins (CD9, CD63, CD81 и CD82), heat shock proteins, Tsg101, ESCRT-3, Alix, etc. [7]. In our study, several markers of plasma EVs were analyzed: CD9, EpCAM (transmembrane glycoprotein), and flotillin-1 (cytosolic protein with membrane-binding ability). All these proteins were revealed in EVs concentrated samples using Western blot analysis (Figure 2). EVs concentrated samples after the first treatment with detergent Triton X 100, proteinase K, and RNAse A did not show any bands after immunoblotting, since all the protein molecules such as flotillin, CD9, and EpCAN were destroyed by proteinase K (Figure 2, column 1). A second treatment with proteinase K and RNAse A degraded only the membrane-associated CD9 and EpCAM molecules, but not flotillin-1 localized within the vesicles, and biotillin-1 was determined after blotting (Figure 2, column 2). The third untreated samples demonstrated bands corresponding to all three proteins (CD9, EpCAM, flotillin-1) (Figure 2, column 3).

### 2.4. Low-Voltage Scanning Electron Microscopy

Differently treated samples of EVs were characterized using low-voltage scanning electron microscopy (LVSEM) (Figure 3). Control slides with DPBS solution contained only single salt crystals but not any spherical objects (Figure 3b). Untreated samples were enriched with protein complexes that covered near-spherical objects (EVs) from 150 to 500 nm in diameter (Figure 3e,f). The treatment with proteinase K and RNAse A resulted in the almost complete disappearance of protein aggregates. At the same time, near-spherical objects on the slides were numerous and well visualized. Their sizes varied from 70 to 300 nm and were smaller than those in the untreated samples slides. Aggregates were sporadic (Figure 3c,d).

Samples treated with Triton X 100 demonstrated absence of organized structures or objects, particularly near-spherical EVs. Slides were enriched with anamorphic debris, salt crystals, and protein fragments (Figure 3a).

### 2.5. Dynamic Light Scattering

The objects in the untreated samples were mainly represented by particles with size specific for vesicles; in this case, the apparent hydrodynamic radii of the particles were 30 and 250 nm, respectively (Figure 4a). After the treatment with proteinase K and RNAse A, the tested samples contained particles of the same size as an apparent hydrodynamic radius of about 130 nm (Figure 4b). After further treatment with Triton X 100, in comparison with the sample, presented in Figure 4b, we observed a significant decrease in the concentration of particles with an apparent hydrodynamic radius of about 120–130 nm (Figure 4c), while the intensity of light scattering significantly decreased, and its optimal values had to be corrected by varying intensity of the laser beam to minimize the size measurement error. In addition, we detected micron-sized objects with hydrodynamic radius of about 800 nm, apparently representing aggregates of broken parts of vesicles (Figure 4c).

### 2.6. Nanoparticle Tracking Analysis

According to NTA data, the concentration of EVs did not differ in untreated and treated with proteinase K and RNAse A samples, whereas the concentration of the objects in the Triton X 100-treated samples decreased compared to Triton X-free samples (Figure 5a). The EV size in untreated samples was larger than in the treated ones (122.0 ± 14.0 nm and 96.3 ± 1.5 nm, respectively).

Particles in the untreated samples appeared with one peak of objects with relatively low intensity (Figure 5b). Two fractions of objects were observed after proteinase K and RNAse A treatments of the samples. An additional peak was presented as objects of low size with high intensity (Figure 5c). Treatment of the samples with Triton X 100 resulted in the appearance of a number of objects of predominantly small size and with various intensity (Figure 5d).

### 2.7. High-Sensitivity Flow Cytometry

Platelet and erythrocyte markers were chosen for immunophenotyping of EVs because platelet-derived and erythrocyte-derived vesicles were the most abundant in plasma. In the untreated samples, CD41+ EVs were detected in the serial dilutions, demonstrating the optimal ration as 1:100. All the events were lysed performing detergent+ control, indicating the membrane-containing structure of the objects (Figure 6, upper row). Erythrocyte-originated CD235a+ EVs were more abundant, making the optimal dilution to avoid coincidence effect 1:1000. Almost all events were lysed after treatment of stained vesicles with Triton X 100 performing membrane-containing structure of the objects (Figure 6, bottom row).

The treatment of the samples with proteinase K and RNAse A resulted in the decrease of CD41+ and especially CD235a+ events, leading to the necessity to use only 1:100 dilution of the samples (Figure 7a, upper row), whereas the remaining events demonstrated isolated staining, with one of the markers without double-positive staining (Figure 7b, upper). The treatment of the samples with proteinase K, RNAse A, and Triton X 100 eliminated almost all CD235a+ events and most CD41+ events. It led to an increase in the level of debris (Figure 7a, bottom row). The presence of double-positive stained events pointed to antibodies-containing aggregates (Figure 7b, bottom).

Identification of the tetraspanins in the structure of the objects is one of the main criteria indicating the presence of EVs in the samples. Thus, we immunophenotyped samples using anti-tetraspanins antibodies (CD9, CD63, CD81, CD82). The number of CD9+ events was predominant among all tetraspanin markers. To avoid an effect of coincidence, we used 1:1000 or 1:10,000 dilution. Events, positive with other markers, were limited and their amounts were not significant (Figure 8). The use of Triton X 100 resulted in the total destruction of all positive events, primarily CD9+. These data confirmed the presence of the phospholipid membrane in the structure of the objects (Figure 8, rightmost column).

As result of treatment of the samples with proteinase K and RNAse A, all objects became negative with tetraspanin markers (Figure 9, upper row). Further processing with Triton X 100 resulted in the significant increase of unclassified debris (Figure 9, bottom row).

### 2.8. The Absolute Number of miRNA Molecules in the EVs Samples Obtained from 1 mL of Plasma after Enzymatic Treatments

In this study, we used TaqMan MiRNA assays as a highly sensitive and highly specific system of miRNAs detection (ThermoFisher scientific, Waltham, MA, USA) [49]. The normalization by volume, concentrating EVs from 1 mL of plasma, and exogenous reference spike-in controls were used.

First, we determined the absolute number of miRNA molecules in the samples after different enzymatic treatments. Previously, it was established that during differential centrifugation, extracellular nonvesicular miRNAs can partially coprecipitate with the EV population, adhering externally to the bilipid membrane, as well as with large plasma proteins [50]. Therefore, in order to exactly quantify intraluminal miRNAs, we applied enzymatic treatments (Figure 1). Using combined sequential proteinase K and RNAse A treatment, we degraded miRNAs that coprecipitated outside EVs and revealed only intraluminal miRNA. We determined the absolute amounts of miRNAs of interest derived from 1 mL of human plasma by using RT-qPCR with standard curve performed with synthetic oligonucleotides of known concentration that were completely identical to the miRNAs under study. We calculated the absolute number of molecules in the standard samples using Avogadro’s number and by interpolation and found the absolute value of the miRNAs of interest in the studied samples. The data are presented in Figure 10.

### 2.9. The Absolute Number of miRNA Molecules in the EVs Samples and Adhered Outside the EVs

We calculated how many molecules of the tested miRNAs were located inside EVs and how many coprecipitated outside the vesicles. After treating the samples with Triton X 100, Proteinase K, and RNAse A, we found that not all miRNAs were completely removed from the samples as might be expected (Figure 10 and Figure 11). The remaining miRNAs were a mixture of undegraded molecules, both coprecipitated nonvesicular and vesicular molecules, which we could not quantitatively separate in this study. We counted the number of vesicular and coprecipitated EVs molecules presented in Figure 12a as follows. Assuming that the efficiency of the enzymes in the tested samples was the same, we used the number of miRNAs after the three treatments as the “error”. Subtracting the “error” value from the number of molecules after treatment with proteinase K and RNAase A, we calculated the number of miRNA molecules inside the vesicles (Figure 12). We then sequentially subtracted the “error” value and the number of molecules inside EVs from the number of miRNA molecules in the untreated samples and calculated the number of miRNA molecules coprecipitated with the EVs population outside the vesicles (Figure 12). The number of miRNAs inside the vesicles was up to twofold lower than those coprecipitated on the outside surface, whereas the ratio of the Let 7f molecules inside/outside the vesicles was 2.64 (Figure 12a).

## 3. Discussion

Blood-derived plasma and serum are the most commonly studied biofluids for extracellular vesicles research [51]. On the other hand, blood is a complex fluid rich in soluble macromolecules (including lipoproteins), cells (intact and fragmented), and nonvesicular nucleic acids that can strongly overlap with EVs in physical properties and composition [51]. Thus, obtaining a pure EVs preparation, in particular from blood plasma, remains an unachieved goal [51]. Nevertheless, it is very challenging to separate EVs from blood plasma or serum to perform a study using one of the available isolation methods [7]. According to the recovery and specificity of achieving fraction, the current recommendations of the International Society for Extracellular Vesicles (MISEV2018) classifies all the methods for EV isolation into four groups [7]: (1) High recovery, low specificity (e.g., precipitation kits, low-molecular-weight cut-off centrifugal filters, very-high-speed ultracentrifugation without previous, lower-speed steps, etc.); (2) Intermediate recovery, intermediate specificity (e.g., size-exclusion chromatography, differential ultracentrifugation using intermediate time/speed with or without wash, tangential flow filtration, membrane-affinity columns etc.); (3) Low recovery, high specificity (immuno- or other affinity isolation including flow cytometry for large particles etc.); (4) High recovery, high specificity (may not be achievable as of the writing of the recommendations) [7]. In the present study, we employed the method of EVs isolation with initial consequential steps of centrifugation of fresh nonfreeze plasma and following 0.8 µm filtering to eliminate platelets, fragmented platelets, and erythrocytes as a major confounders of EV preparation [51,52]. In addition, we performed the high-speed centrifugation followed by washing procedures. Thus, according to MISEV2018 recommendations, the used method may be qualified as intermediate recovery, intermediate specificity, allowing the recovery of mixed EVs along with some amount of free proteins and lipoproteins [7].

MiRNAs are known to play an important role in pathogenesis of numerous pathological conditions and illnesses [53,54,55,56]. These small objects are very effective participants in the regulation of many cell processes as well as intercellular communication in physiological and pathological conditions [54,57]. Thus, their easy transfer from cell to cell is one of the main conditions to realize their functions. Nucleic acids need to be protected from the action of RNAse because of their accessibility to the destructive activity of enzymes. The MiRNA binding protein Ago2 is able to act as a protector for nucleic acids and, thus, participate in realization of miRNA functions [29,48,58]. In addition, easy transport between cells from the different tissues is also a very important criterion for miRNA-dependent intercellular communication and cell activity regulation.

EVs, generated inside the cells via stage of multivesicular bodies (exosomes) or blabbing from the outer cell membrane (microparticles), are able to load the cargo from host cell and transfer it to the recipient cell [59]. Extracellular vesicles include DNA fragments, mRNA, miRNA, enzymes, signaling molecules, cytoskeleton proteins, cell-type specific proteins, tetraspanins etc. [59]. Thus, the miRNA, located in the inner space of the vesicle, is supposed to be produced and packed by the host cell. Extracellular vesicles are a useful instrument for targeted delivery of the cargo to closely or distantly located cells, playing the role of destination-oriented vehicle. In addition, EVs demonstrate protection abilities for transported content, serving as reliable container. Proteins and lipids allow the protection of nucleic acids from the enzymes in the same way as Ago2 does. Thus, EVs are assumed to be a useful reliable container on the destination-oriented vehicle for transit of the nonhost-cell-produced miRNA.

How many miRNA can fit in a single EV, taking into account the inner vesicle space and outer membrane surface? Previously, we theoretically calculated the possible number of miRNAs packed inside a single vesicle [18]. Assuming that miRNAs inside the vesicle is presented as a protein complex [48], we can approximate it as a sphere with diameter equal to miRNA length. If the length of a single RNA nucleotide is 0.34 nm and miRNA consists of 22 nucleotides, the miRNA length should be ~7.5 nm and the volume calculated using the formula V = 4πr^3^/3 should be ~221 nm^3^. For a spheroid vesicle of 100 nm in diameter and 4–5 nm wall thickness [60], the volume of a vesicle should be approximately ~382,000 nm^3^. Thus, the maximum number of miRNAs that can be placed in a vesicle depends on the packing efficiency of equal spheres in three-dimensional space. In the case of random packing, the density is 64% of the filled volume [61]. Thus, the available volume of the 100 nm vesicle is equal to ~244,000 nm^3^ and its capacity is more than 1000 miRNA–protein complexes [18]. In these calculations, we ignore that vesicles are fitted not only with miRNA. In addition, we are not taking into account that hydrophilic amino acids on the surface of proteins, miRNA nucleotides, and hydrophilic lipid heads are hydrated by water molecules, and this will also affect the possibility of miRNA packaging in a vesicle. Nevertheless, we can think of the number of molecules that are potentially transported inside a single EV. In addition, previously, we experimentally demonstrated that the number of miR-223-3p and miR199a-3p in the CD41+CD235a− EV was 38.2 and 2.8 molecules per event, respectively, whereas the numbers of miR-451a in the CD41-CD235a+ EVs and in the CD41-CD235a dim EVs were 24.7 and 3.2 molecules per event, respectively [18].

The classical sphere-packing problem, still unsolved even today, is to find out how densely a large number of identical spheres can be packed together [61]. This problem is also known as “contact number” or “Newton’s number”. The number of identical spheres to be packed in the region in 3D space contacting with one sphere is 12. Thus, we assume that, in the case that miRNAs with 8 nm diameter each are adhering on the surface of the EV that has a 100 nm diameter, their density could be much greater than in identical-volume spheres. It demonstrates the ability of EVs to transport, or at least protect, an extremely high number of miRNAs.

There are limited papers investigating the ratio of miRNAs per EVs, especially in pathological conditions. Chevillet and colleagues showed that even abundant miRNAs are present in amounts much lower than one copy per exosome even in physiological conditions [62]. In addition, the numbers of copies of miRNA per exosome for miRNA-126 and miRNA-223 in patients with prostate cancer were higher than those in healthy donors [62]. We suggest that the number of miRNAs transported by EVs can be significant in the pathogenesis of various diseases.

Our results show that the average number of miRNA molecules (hsa-let-7a, hsa-let-7f, hsa-30a, hsa-486, hsa-223, hsa-451) inside the vesicles was slightly higher than those outside (Figure 11). Variations in the inside/outside ratios of the number of molecules ranged from 1.02 to 2.64 (Figure 11A), knowing that the number outside miRNA could be more than 12-fold higher then inside. Thus, coprecipitated and externally adhered RNA molecules contribute to the quantification of RNA of interest, and it needs to be considered if the samples are not treated with proteinase and RNAse. Probably, the presence of transit, nonhost-cell-produced miRNAs on outer surface of EVs explains previous results, demonstrating significant similarities in RNA repertoire between plasma vesicles of different cell origin [17]. Meanwhile significant differences are more expected [18,63] because of the original miRNA profile of each cell type, including peripheral blood cells. Previously, Chevillet et al. [62] proposed four stoichiometry models for exosome miRNA content. Authors suggested that high-occupancy/high-miRNA concentration model and high-occupancy/low-miRNA concentration models are not relevant and, thus, should be ignored. However, a low-occupancy/low-miRNA concentration model, in which a small fraction of exosomes carries a low concentration of miRNA, and a low-occupancy/high-miRNA concentration model, in which there are rare exosomes in the population carrying many copies of a given miRNA, are the most reliable [62]. According to our results, we propose another hypothesis of high occupancy of miRNA but on the outer surface of extracellular vesicles influencing the transporting RNA repertoire no less than inner cargo received from the host cell. In addition, a low-occupancy/high-miRNA concentration model with increased number of produced miRNA-free exosomes related to specific cell activity and decreased need for miRNA delivery may serve as a protecting depot for free-circulating RNA. There is no doubt that extracellular vesicles generating activity of the cell depends on their functional status, conditions, and even a producing compartment of each single cell [64].

Analyzing the outside and inside cargo of extracellular vesicles, we focused on concrete miRNAs, not taking into account the heterogeneity of vesicles presenting in plasma. On one hand, we proposed different inside/outside ratios for different molecules; on the other hand, vesicle phenotyping demonstrated presence of different cell origin and different receptor expression profile of EVs in the samples. In addition, NTA-detected intensity, DLS-characterized dispersion, and SEM-described morphology confirmed the statement of heterogeneity of studied samples.

It was reported that miRNAs in plasma were well protected from nucleases [11,32,33]. We also showed that not all miRNAs were destroyed after three treatments (Figure 10, Figure 11 and Figure 12). We rarely detected single vesicles on SEM preparations after triton treatment; perhaps they also contributed to the total pool of remaining miRNAs in the treated sample, or extracellular nonvesicular miRNA is firmly protected by proteins that are not completely removed by proteinase K.

According to these findings, several aspects should be remembered when working with miRNAs and extracellular vesicles from blood plasma. First, the currently existing methodological approaches are imperfect and distort the real picture. Unstable RNAs degrade in the body after leaving the cells and during isolation or sample preparation, only approximately reflecting what is actually present in the sample [65]. Second, plasma is a complex material with a large number of proteins, protein complexes, and lipoproteins masking vesicles [7,66]. These conditions complicate the phenotyping of extracellular vesicles [67]: they make it difficult to bind fluorescent-labeled antibodies, increasing false-negative events ratio or, vice versa, mimic proteins and lipoproteins to EVs, increasing the false positive events ratio. Large amounts of protein were observed in samples without proteinase K treatment for all methods used (Figure 3e,f). Proteinase K treatment of EVs resulted in the protein aggregates’ disappearance and single vesicles’ sample enrichment, but at the same time, EV receptors degraded, making phenotyping impossible (Figure 7 and Figure 9). Thus, while working with plasma extracellular vesicles, the proteinase treatment of the samples can be recommended for pure single vesicle population achievement, but only taking into account limitations, related to partially destructions of vesicles themself and their receptors.

In current study, we analyzed the absolute number of miRNAs (hsa-30a, hsa-223, hsa-let-7a, hsa-let-7f, hsa-451, hsa-486,) inside and attached to the outside of plasma EVs. MiRNA-30a was identified by Lagos-Quintana and colleagues and is widely distributed in humans, accounting for 4.1% of the total number of miRNAs expressed [68]. It originates from an intronic transcriptional unit and is involved in a wide range of biological processes, including cell differentiation and development, and plays a dual role as an oncogene or oncosuppressor in various cancers [42,69]. miRNA 223 is transcribed from an independent promoter and is highly expressed in the hematopoietic system [39,70,71]. MiRNA 223 expression changes considerably during cellular differentiation. It increases during osteoclast, megakaryocyte, and eosinophil differentiation and decreases during erythrocyte and macrophage differentiation [72]. The highly conserved let-7 miRNA family includes, among others, let-7a and let-7f miRNAs, which are clustered together. The let-7 family of miRNAs is expressed during embryogenesis and brain development and is a tumor suppressor [73]. MiRNA let-7a and let-7f were identified by Lagos-Quintana and colleagues and have canonical biogenesis pathway [68]. MiRNA let-7a has three genes in the human genome and is highly expressed in the skin and submandibular gland. miRNA let-7f has two genes in the genome and is highly expressed in the sclera and submandibular gland [74]. miRNA 451 has one gene in the genome and is highly expressed in red blood cells and bones [70]. miRNA 451 has noncanonical Drosha/DGCR8-dependent/Dicer-independent biogenesis [75,76]. MiRNA 486 is transcribed from an intron within the ANK1 locus [70,77]. It has high expression in skeletal and cardiac muscle and erythroid cells and circulates at high levels in plasma [41]. MiRNA 486 is enriched within small extracellular vesicles and has atypical biogenesis [78,79].

These molecules were chosen as the most abundant miRNAs circulating in plasma [17,80]. Moreover, has-let-7hashsa-let-7f, and hsa-486 were not only among the top 20 most abundant molecules for different body fluids, but were among 5 miRNAs common to all fluids (blood, saliva, urine, plasma, serum) [81]. These five miRNAs represent more than 50% of blood and serum miRNA counts, 25 to 45% of plasma and saliva [81]. Platelet-derived and erythrocyte-derived EVs are the main populations of vesicles in a blood plasma [82]. Therefore, hsa-451, known to be an erythrocyte-enriched miRNA [63,81,83] and an erythrocyte-derived extracellular vesicle-specific [18,63] marker, and hsa-223, the most abundant miRNA in megakaryocytes and platelets [84,85], were additionally chosen for the study.

Thus, investigating the most abundant miRNAs in plasma and most specific miRNAs for the most abundant plasma extracellular vesicles, we demonstrated that each type of molecule is transported inside the vesicles as well as on the outside surface. We assume that different miRNAs are allowed to adhere on the outer membrane of EVs, but this ability varies depending significantly on the physical–chemical peculiarities of concrete miRNA and vesicles subpopulations.

## 4. Materials and Methods

### 4.1. Study Population

This study was performed according to the Helsinki Declaration and was approved by the local ethics committee (protocol No. 2209-20, 21 September 2020) of the Almazov National Medical Research Centre. All participants provided written informed consent. The analysis included 12 healthy male volunteers from 18 to 40 years old. Blood samples were taken from the peripheral vein.

### 4.2. Sample Collection and Preparation

Whole blood from peripheral vein was collected in Vacutainer tubes with EDTA (Vacutest, KIMA, Arzergrande, Italy). In order to minimize the effect of EVs release from blood cells, blood samples were processed within 1 h after collection. In the first step, venous blood samples were centrifuged two times successively at 1500× *g* for 10 min at +18 °C and one time at 3000× *g* for 30 min at +4 °C using a swinging bucket rotor. After each centrifugation step, the supernatant was carefully transferred to a new conical tube. To avoid getting platelets in the samples the supernatant was filtered through a 0.8 µm filter (Merck Millipore, Darmstadt, Germany). In the second step, precipitate of EVs was obtained from 1 mL of supernatant by centrifugation at 20,000× *g* for 2 h. The resulting pellet was washed with 1 mL Dulbecco’s PBS (DPBS) (Biolot, St. Petersburg, Russia) and stored at −80 °C for further use.

### 4.3. Enzymatic Treatment

Three EV concentrate samples were obtained from each donor. Every concentrate was prepared from 1 mL of previously treated plasma. First concentrate samples from each donor were treated as follows: 1% Triton X 100 (Sigma-Aldrich, St. Louis, MO, USA) for 30 min at RT, proteinase K (Sigma-Aldrich, St. Louis, MO, USA) (1 mg/mL) for 2 h at 60 °C, then proteinase K inactivation by PMSF (Merck Millipore, Darmstadt, Germany) (1 mM) for 30 min at RT, followed by RNAse A (Sigma-Aldrich, St. Louis, MO, USA) (100 µg/mL) treatment for 1 h at 37 °C. Second concentrate samples from each donor were treated as follows: proteinase K (Sigma-Aldrich, St. Louis, MO, USA) (1 mg/mL) for 2 h at 60 °C, then proteinase K inactivation by PMSF (Merck Millipore, Darmstadt, Germany) (1 mM) for 30 min at RT, followed by RNAse A (Sigma-Aldrich, St. Louis, MO, USA) (100 µg/mL) treatment for 1 h at 37 °C. In the third concentrate samples from each donor, only an equal volume of DPBS was added, followed by incubation at the same temperatures as the enzymatic treatments. Immediately, Trizol LS (Life Technologies Co., Carlsbad, CA, USA) was added to the samples that were subsequently used for RNA isolation (description in the section below).

### 4.4. Western Blot Analysis

The samples of equivalent volume were denatured for 5 min at 95 °C in Laemmli sample buffer (Bio-Rad, Hercules, CA, USA) containing 2-mercaptoethanol (Bio-Rad, Hercules, CA, USA). The protein separation was carried out in 10–12.5% gradient polyacrylamide gels (Bio-Rad, Hercules, CA, USA) under reducing (anti-flotillin-1, anti-EpCAM) or nonreducing (anti-CD9) conditions at 150 V. After that, proteins were transferred to a nitrocellulose membrane (Bio-Rad, Hercules, CA, USA) with subsequent Ponceau S staining (Thermo Fisher Scientific, Waltham, MA, USA). The PageRulerTM (Thermo Fisher Scientific, Waltham, MA, USA) was used as a molecular weight standard. Then, the membranes were blocked in 5% (*w*/*v*) skim milk powder in 1× PBS buffer with 0.1% Tween-20 for 90 min and incubated overnight at 4 °C with the primary antibodies (dilution 1:1000): flotillin-1 (D2V7J) XP rabbit mAb 18634 (Cell Signaling Technology, Danvers, MA, USA), CD9 (D8O1A) rabbit mAb 13174 (Cell Signaling Technology, Danvers, MA, USA), and EpCAM (D6B3) rabbit mAb 2626 (Cell Signaling Technology, Danvers, MA, USA). After washing three times in TBS buffer with 0.1% Tween-20, the membranes were incubated with secondary anti-rabbit IgG, HRP-linked Antibody 7074 (Cell Signaling Technology, Danvers, MA, USA) for 1 h at room temperature. Visualization was performed with SuperSignal West Femto Maximum Sensitivity Substrate (Thermo Fisher Scientific, Waltham, MA, USA). Imaging was performed using a Fusion FX Vilber Lourmat gel documentation system (Vilber, Collégien, France).

### 4.5. Low-Voltage Scanning Electron Microscopy

Samples were visualized using low-voltage scanning electron microscopy (LVSEM) as described previously [29,86,87]. Menzel microscope glass slides (Thermo Fisher Scientific, Waltham, MA, USA) washed with 2% 7X cleaning solution (Helicon, Moscow, Russia) were used. Briefly, 5 µL of each EVs concentrate in DPBS were placed on the slides and fixed in equal volume of 5% glutaraldehyde solution on PBS during 30 min in a humidified atmosphere. After that, samples were dehydrated in a series of ethanol solutions and air dried.

Slides were analyzed on a Zeiss Merlin scanning electron microscope with auto-emission cathode (Carl Zeiss, Oberkochen, Germany) at an accelerating voltage was of 200–400 V and beam current of 80–100 pA. The pressure in the microscope chamber was about 7 × 10^−5^ Pa.

### 4.6. Dynamic Light Scattering

The apparent hydrodynamic size and size distribution of objects in the samples were determined by dynamic light scattering (DLS). The measurements were performed on a Photocor Compact Z instrument (Photocor Ltd., Moscow, Russia) with a 659.7 nm He–Ne laser at 25 mV power, a detection angle of 90°, and at a temperature of 20 °C. The sample equilibration time was not less than 60 s. Each sample was measured in triplicate. Pure DPBS was used as a reference solution and sample purity control. DynaLS software V1.0 (Photocor Ltd., Moscow, Russia) was used for data analysis.

### 4.7. Nanoparticle Tracking Analysis

The size and concentration of the particles were investigated using nanoparticle tracking analysis technology on NanoSight NS300 (Malvern Panalytical, Malvern, UK). Samples were diluted in ultrapure water (Merck Millipore, Darmstadt, Germany) to obtain a concentration within the manufacture’s recommended range. Video capture was performed in laminar streaming mode of the sample. The settings for analysis were as follows: detection threshold 5; temperature between 20 °C and 23 °C; measurement time 60 s; and number of frames per second—25. Each sample was measured in 5 technical replicates. Particle size distribution and concentration was presented as the average of five independent measurements. NTA NanoSight V.3.4 (NanoSight Ltd., Malvern, UK) software was used for the results processing.

### 4.8. High-Sensitivity Flow Cytometry

Immunofluorescence staining of the samples was carried out using the following antibodies (all antibodies were obtained from BioLegend Inc., San Diego, CA, USA): anti-CD41-AF488 (clone: HIP8, cat.303724, concentration 100 µg/mL) and anti-CD235a-PE (clone: HI264, cat.349106, concentration 0.2 µg/mL) as platelet- and erythrocyte-specific markers, respectively; anti-CD9-Pe/Cy7 (clone: HI9a, cat.312116, concentration 50 µg/mL), anti-CD63-APC (clone: H5C6, cat.353008, concentration 200 µg/mL), anti-CD81-FITC (clone: 5A6, cat.349504, concentration 200 µg/mL) and anti-CD82-PE (clone: ASL-24, cat.342104, concentration 200 µg/mL) as markers of tetraspanins. The first two antibodies were chosen as markers of the most numerous plasma EVs subpopulations (platelet- and erythrocyte-derived), whereas anti-tetraspanins antibodies identify EVs.

Samples’ phenotypes were investigated using two panels of antibodies performed separately in appropriate tubes. First, 50 µL of each sample was stained with 0.5 µL of each monoclonal antibody (anti-CD41 and anti-CD235a or tetraspanins, respectively) during 25 min at 20 °C in the dark. Isotype controls—Alexa Fluor 488 Mouse IgG1, κ Isotype Ctrl (BioLegend Inc., San Diego, CA, USA, clone: MOPC-21, cat. 400129, concentration 200 µg/mL), PE Mouse IgG1, κ Isotype Ctrl (BioLegend Inc., San Diego, CA, USA, clone: MOPC-21, cat. 400114, concentration 200 µg/mL), APC Mouse IgG1, κ Isotype Ctrl (BioLegend Inc., San Diego, CA, USA, clone: MOPC-21, cat. 400122, concentration 200 µg/mL)—were stained at the same conditions and concentrations as appropriate fluorochrome marked antibodies. Then, 20 µL of each stained sample was diluted tenfold with DPBS. After that, these prediluted samples were used to perform a series of five consecutive tenfold dilutions which were further used to measure EV concentration in the initial material.

A buffer-only control of DPBS (Biolot, St. Petersburg, Russia) and a buffer with reagent controls (0.5 µL of each monoclonal antibody in 50 µL of DPBS) were recorded at the same flow cytometer acquisition settings as all samples, including triggering threshold, voltages, and flow rate. All controls used were diluted 5 times with DPBS in the same way as the samples to allow comparisons between serial diluted stained samples. Buffer with reagent control had the same event rate as buffer-only control.

Unstained controls were measured at the same dilution as stained samples and isotype controls. Flow cytometer acquisition settings for all these controls were unchanged, including triggering threshold, voltages, and flow rate. No substantial changes were observed in scatter or fluorescence signals between unstained and matched isotype controls.

Samples were serially diluted five times. Dilutions of 1:1000, 1:10,000, and 1:100,000 demonstrated a linear correlation between dilution factor and measured event count, whereas the fluorescence of the events and scatter intensity did not change. Thus, the dilution of 1:1000 was chosen to calculate EVs concentration and to present the results of immunostaining and EVs phenotyping.

Phenotyping analysis of EVs was performed using a Cytoflex S (Beckman Coulter, Indianapolis, Indiana, USA) flow cytometer equipped with 405 nm (violet), 488 nm (blue), 638 nm (red), and 561 nm (yellow–green) lasers. Detection was triggered on a 405 nm laser at the threshold of 2000 arbitrary units. Instrument calibration setup was performed using the Cytometry Sub-Micron Particle Size Reference Kit, Molecular probes by Life Technologies, Carlsbad, CA, USA), and Megamix-Plus FSC and Megamix-Plus SSC (Biocytex, Marseille, France) containing FITC-labeled reference beads of various diameters as described previously [13,18,86]. Samples were enumerated using the integral flow rate sensors. The sample flow rate was set to 120 µL min^−1^. Stained objects were detected using side scatter from the violet 405 nm laser and appropriate to the used fluorochrome-labeled antibodies light pass channel.

The data obtained were analyzed using CytExpert v.2.4 (Beckman Coulter, Indianapolis, IN, USA) and Kaluza 2.1 (Beckman Coulter, Indianapolis, IN, USA) software.

### 4.9. Isolation of Total RNA

Total RNA was isolated from EVs concentrate by phenol-chloroform extraction using the TRIzol LS reagent (Life Technologies Co., Carlsbad, CA, USA) as described in Kondratov et al. 2020 [18]. Briefly, 100 µL of EVs concentrate was mixed with 300 µL of TRIzol LS containing 1 µg of *Escherichia coli* tRNA (# R1753-100UN, Sigma-Aldrich, St. Louis, MO, USA) and 108 molecules of Caenorhabditis elegans synthetic oligoribonucleotide synth-cel-miR-39-3p (UCACCGGGUGUAAAUCAGCUUG) (Syntol, Moscow, Russia) identical to mature cel-miR-39-3p (miRBase: MIMAT000001). Synthetic oligoribonucleotide cel-miR-39-3p was used as an exogenous control to check the quality of miRNA isolation. GlycoBlue™ was used as coprecipitant (Thermo Fisher Scientific, Waltham, MA, USA). The dried RNA precipitate was dissolved in 12 μL of RNase-free water (Ambion, Austin, TX, USA). The samples were stored in the freezer at −80 °C.

### 4.10. Reverse Transcription and Real-Time Polymerase Chain Reaction

In this work, expression levels of the following miRNAs were analyzed: hsa-miR-let-7a-5p MIMAT0000062, hsa-miR-let-7f-5p MIMAT0000067, hsa-miR-30a-5p MIMAT0000087, hsa-miR-486-5p, MIMAT0002177, hsa-miR-223-3p MIMAT0000280, and mmu-miR-451 MIMAT0001632 (https://www.mirbase.org/ accessed on 20 February 2023).

As the total RNA was isolated from microvesicle concentrate samples prepared from 1 mL of plasma and diluted in the equal volume of water (12 µL), subsequent RNA equalization was not required.

The TaqMan™ MicroRNA Reverse Transcription Kit (Thermo Fisher Scientific, Waltham, MA, USA) was used for reverse transcription according to the manufacturer’s recommendations. The following TaqMan MicroRNA Assays were used to detect miRNAs of interest: hsa-miR-let-7a (Assay ID 000382), hsa-miR-let-7f (Assay ID 000377), hsa-miR-30a (Assay ID 000417), hsa-miR-486 (Assay ID 001278), hsa-miR-223 (Assay ID 002295), mmu-miR-451 (Assay ID 001141), and cel-miR-39 (Assay ID 000200) (Life Technologies, Carlsbad, CA, USA). Reverse transcription was performed in a Veriti 96-Well Thermal cycler (Applied Biosystems, Waltham, MA, USA) with the following amplification program: 30 min at 16 °C, 30 min at 42 °C, and 5 min at 85 °C. After reverse transcription, the cDNA was stored at −20 °C until the amplification.

Amplification of miRNA targets was performed using a TaqMan Universal Master Mix II no UNG (Thermo Fisher Scientific, Waltham, MA, USA) and hydrolysis probes of TaqMan MicroRNA Assays (Thermo Fisher Scientific, Waltham, MA, USA) according to the manufacturer’s recommendations. MiRNA qPCR amplification was performed using a LightCycler 480 II (Roche, Basel, Switzerland) with the following program: 95 °C for 10 min, followed by 40 cycles of 95 °C for 15 s and 60 °C for 1 min. Calibration curves were constructed using a series of 10-fold dilutions of synthetic RNA oligonucleotides (Syntol, Moscow, Russia), identical to mature target miRNAs.

#### Standard Curve

The standard curve method with synthetical oligoribonucleotides (DNA-synthesis, Russia) identical to mature miRNAs (has-miR-let-7a, has-miR-let-7f, has-miR-30a, has-miR-486, has-miR-223, has-miR-451) was used to determine the absolute number of miRNAs of interest in the samples. Detection was applied as described in the previous section.

### 4.11. RT-qPCR Data Analysis

Raw data from thermocycler software were analyzed using GraphPad Prism 8 software Version 8.0.2 (San Diego, CA, USA). The statistical Mann–Whitney U-test was used to assess differences between the two groups. Relative quantification was performed using the 2^−ΔΔCt^ method. Statistically significant *p*-value was <0.05 (* *p* < 0.05, ** *p* < 0.01, *** *p* < 0.001).

#### EV-TRACK

We have submitted all relevant data of our experiments to the EV-TRACK knowledgebase (EV-TRACK ID: EV231001) [88].

## Figures and Tables

**Figure 1 ncrna-10-00015-f001:**
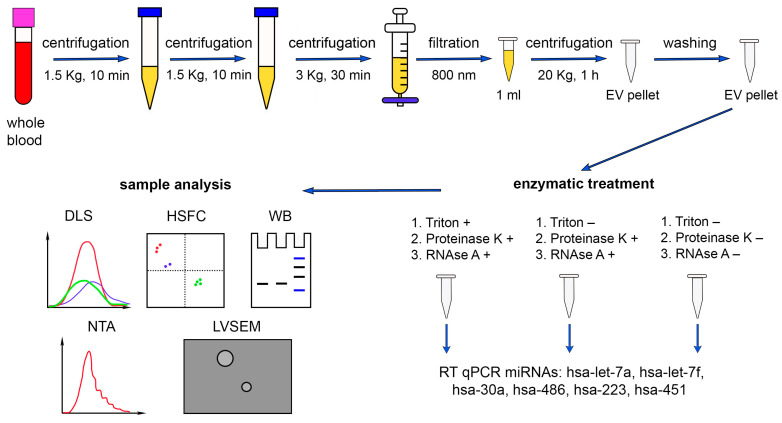
Schematic representation of the miRNA quantitative analysis (hsa-miR-let-7a, hsa-miR-let-7f, hsa-miR-30a, hsa-miR-486, hsa-miR-223, and hsa-miR-451) in human plasma. DLS—dynamic light scattering, HSFC—high-sensitivity flow cytometry, WB—Western blot, NTA—nanoparticle tracking analysis, LVSEM—low-voltage scanning electron microscopy. The results of DLS analysis of the samples after different treatments were overlaid for comparison purposes.

**Figure 2 ncrna-10-00015-f002:**
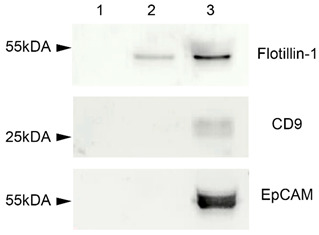
Western blot for flotillin-1, CD9, and EpCAM. 1—EVs concentrated samples treated with detergent Triton X 100, proteinase K, and RNAse A. 2—EVs concentrated samples treated with proteinase K and RNAse A. 3—untreated EVs concentrated samples.

**Figure 3 ncrna-10-00015-f003:**
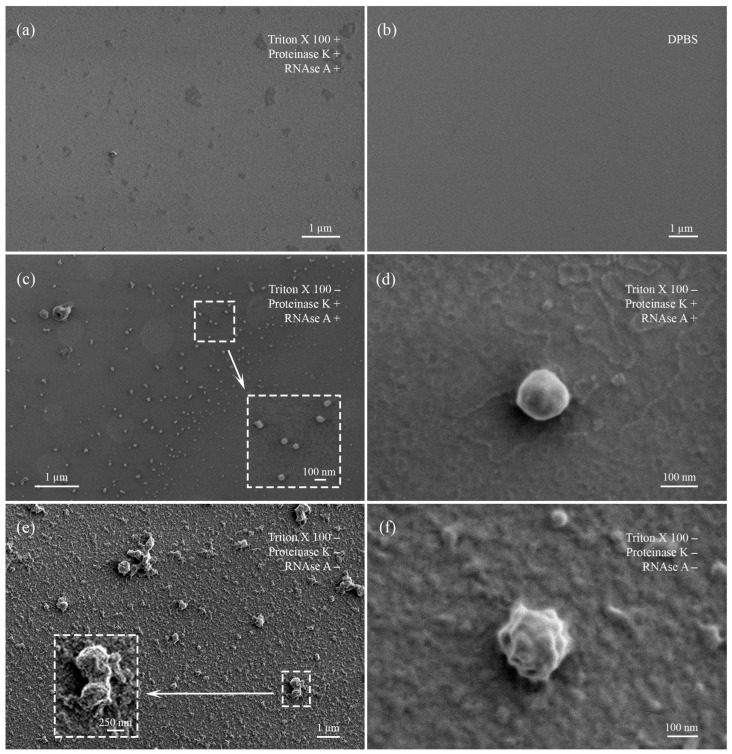
Visualization of treated and untreated EV samples obtained by low-voltage scanning electron microscopy. (**a**) EV sample sequentially treated with Triton X 100, proteinase K, and RNAse. (**b**) Control slide with DPBS. (**c**) EV sample treated with proteinase K and RNAase. Enlarged field marked with dotted rectangle. (**d**) Enlarged single EV, demonstrating absence of protein corona on the outer surface. (**e**) Untreated EV sample. Enlarged field marked with dotted rectangle. (**f**) Enlarged single EV, demonstrating presence of protein corona on the outer surface.

**Figure 4 ncrna-10-00015-f004:**
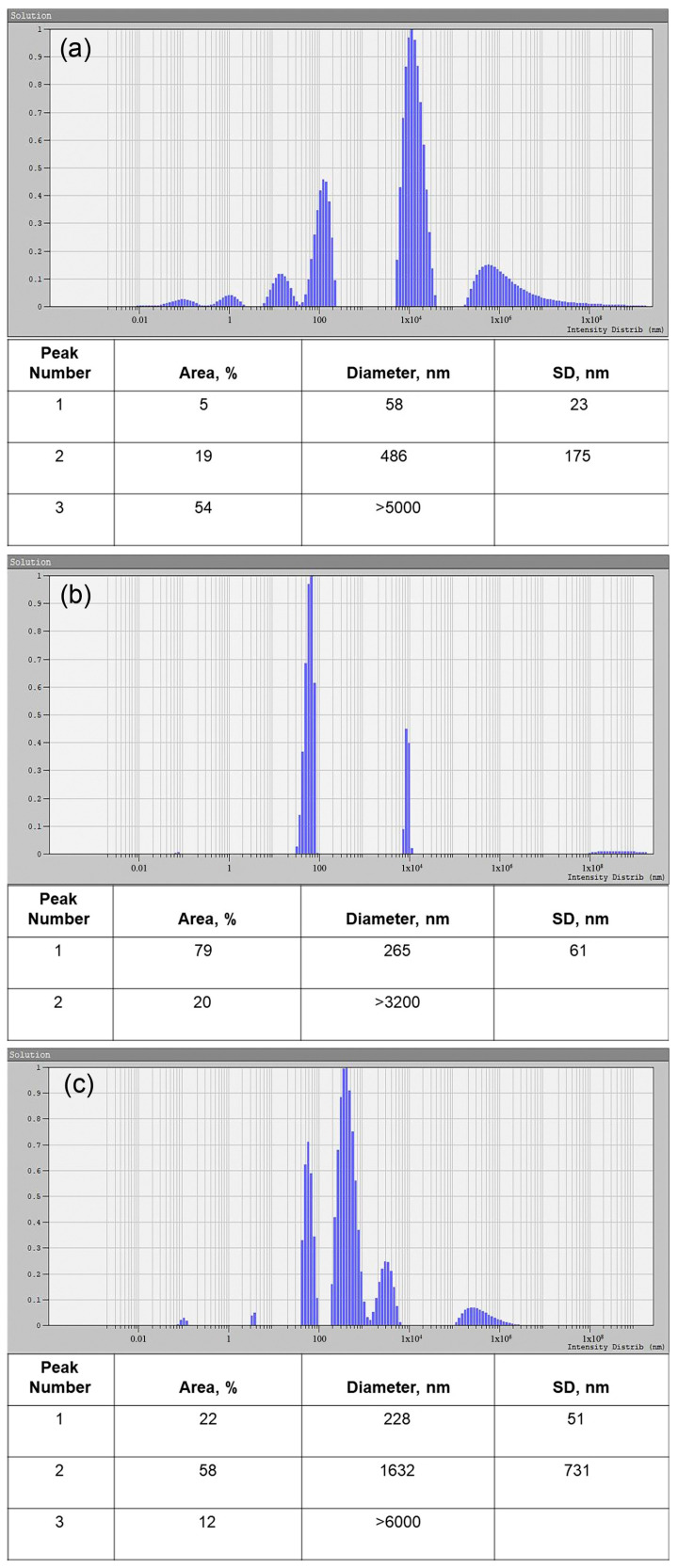
Size distribution of the treated and untreated EV samples, obtained by DLS. (**a**) Untreated EV sample. (**b**) EV sample treated with proteinase K and RNAse A. (**c**) EV sample treated with Triton X 100, proteinase K, and RNAse A. Peak number, area, particle diameter, and standard deviation (SD) are presented in the corresponding tables.

**Figure 5 ncrna-10-00015-f005:**
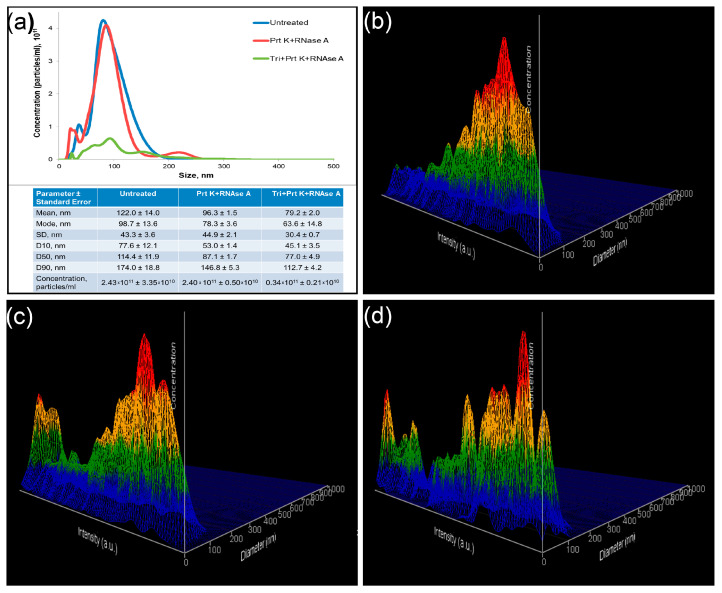
Nanoparticle tracking analysis of treated and untreated samples. (**a**) Comparative analysis of representative measurements of untreated (blue), treated with Proteinase K and RNAse A (red), and treated with Triton X 100, Proteinase K, and RNAse A (green) samples. Representative 3D graph of size distribution (size vs. intensity vs. concentration) of untreated (**b**), treated with Proteinase K and RNAse A (**c**), and treated with Triton X 100, Proteinase K, and RNAse A (**d**) samples.

**Figure 6 ncrna-10-00015-f006:**
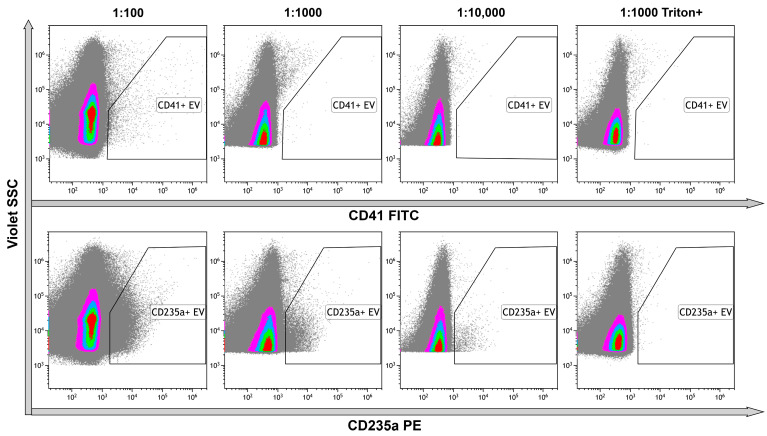
Immunophenotyping of untreated EVs using anti-CD41 and anti-CD235a fluorescent monoclonal antibodies. **Upper row**—representative dot plots of serial dilutions of the sample stained with anti-CD41 antibodies. The last graph on the right illustrates the results of detergent plus control. **Bottom row**—representative dot plots of serial dilutions of the sample stained with anti-CD235a antibodies. The last graphs on the right illustrate the results of detergent plus control. Pseudo-color dot plots reflect relative vesicles’ expression density.

**Figure 7 ncrna-10-00015-f007:**
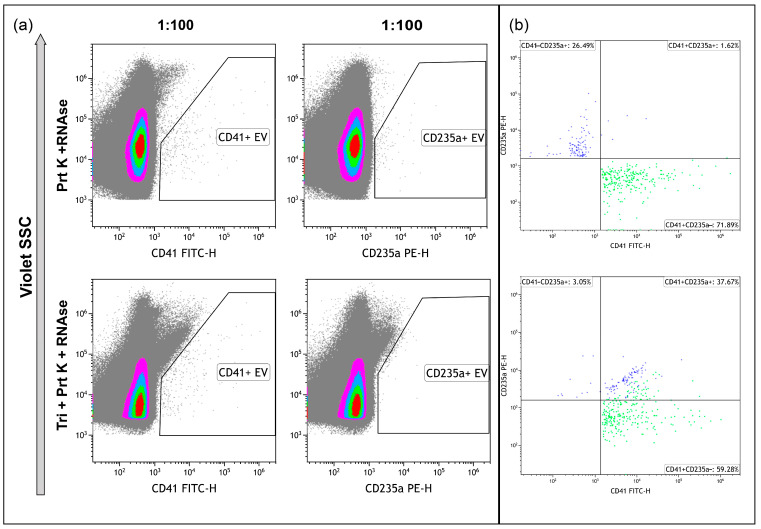
Immunophenotyping of samples, treated with Proteinase K and RNAse A ((**a**), upper row) and treated with Proteinase K, RNAse A, and Triton X 100 ((**a**), bottom row), and stained with anti-CD41 and anti-CD235a antibodies. (**b**) Dot plots representing the results of double-staining of the samples. Pseudo-color dot plots on subfigure (**a**) reflect relative vesicles’ expression density.

**Figure 8 ncrna-10-00015-f008:**
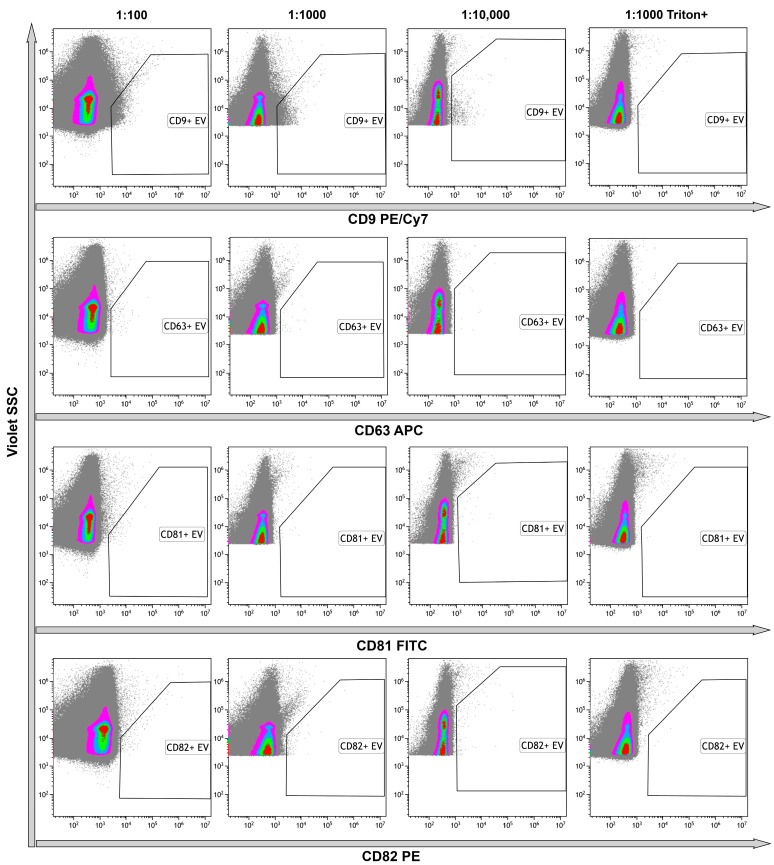
Representative dot plots of the results of untreated samples immunostaining with anti-CD9, anti-CD63, anti-CD81, and anti-CD82 monoclonal fluorescent antibodies. Results presented in serial dilutions. The rightmost column demonstrates the results of detergent plus control. Pseudo-color dot plots reflect relative vesicles expression density.

**Figure 9 ncrna-10-00015-f009:**
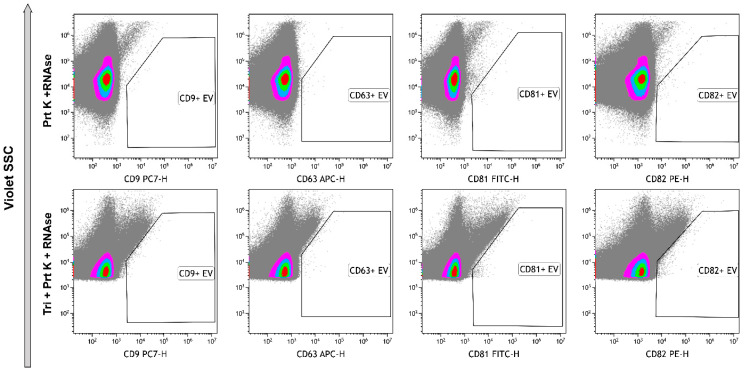
Immunophenotyping of samples, treated with Proteinase K and RNAse A (**upper row**) and treated with Proteinase K, RNAse A, and Triton X 100 (**bottom row**) stained with anti-CD9, anti-CD63, anti-CD81, and anti-CD82 monoclonal fluorescent antibodies. Pseudo-color dot plots reflect relative vesicles’ expression density.

**Figure 10 ncrna-10-00015-f010:**
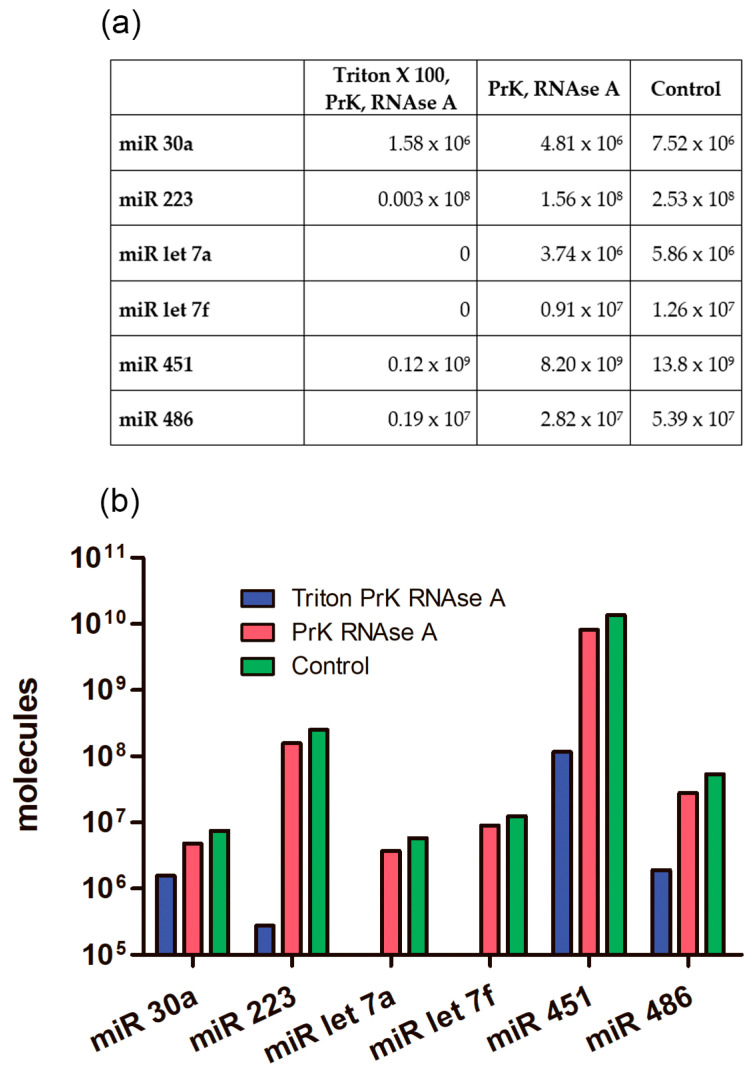
Average number of different miRNA molecules in EV concentrate from 1 mL of plasma after enzymatic treatments. PrK: proteinase K. (**a**) Table view. (**b**) Graphical view.

**Figure 11 ncrna-10-00015-f011:**
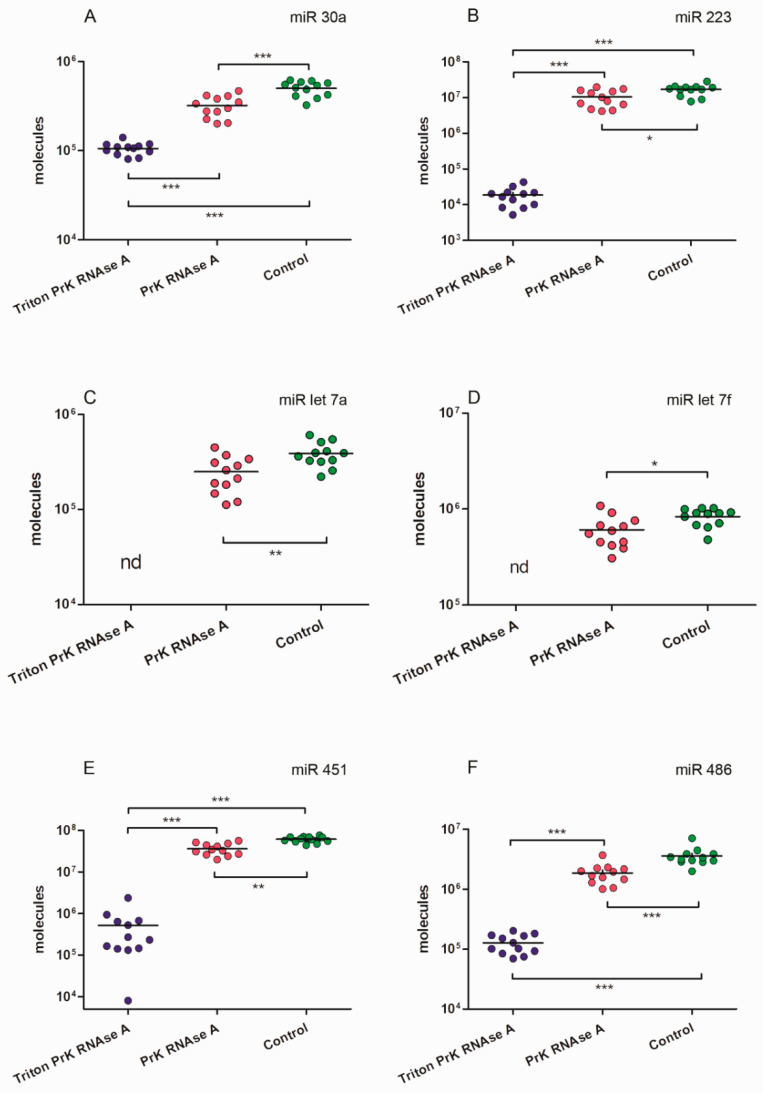
MiRNAs expression levels in samples of EVs concentrate after different enzymatic treatments. (**A**) miR 30a. (**B**) miR 223. (**C**) miR let 7a. (**D**) miR let 7f. (**E**) miR 451. (**F**) miR 486. One asterisk *p* < 0.05; two asterisks *p* < 0.01; three asterisks *p* < 0.001; nd: not determined.

**Figure 12 ncrna-10-00015-f012:**
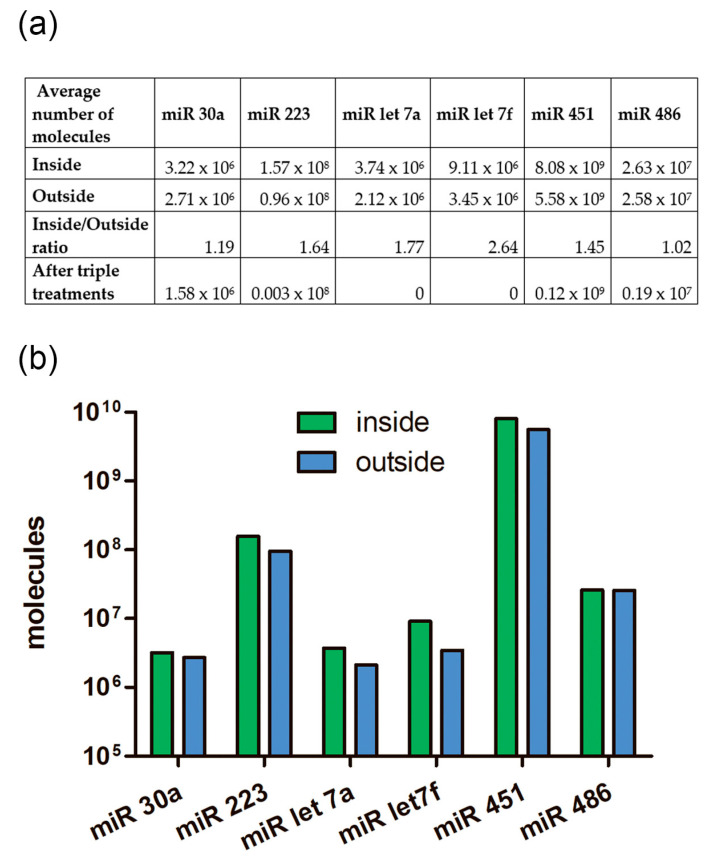
The ratio of the number of miRNA molecules inside and outside of EVs from human blood plasma. (**a**) Table view. (**b**) Graphical view.

## Data Availability

The original contributions presented in the study are included in the article, further inquiries can be directed to the corresponding author.

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
