# Peer review of "Topographic Distribution of miRNAs (miR-30a, miR-223, miR-let-7a, miR-let-7f, miR-451, and miR-486) in the Plasma Extracellular Vesicles"

_ncrna, 2024, doi:10.3390/ncrna10010015_

Round 1

Reviewer 1 Report

Comments and Suggestions for Authors

In the manuscript entitled “Topography distribution of miRNAs (miR-30a, miR-223, 2 miR-let-7a, miR-let-7f, miR-451 and miR-486) in the plasma extracellular vesicle”, Petrova et al. utilized differential centrifugation for the isolation of EVs from human plasma. They conducted comparative enzymatic treatments and employed a suite of analytical techniques including DLS, HSFC, WB, NTA and LVSEM. Subsequently, qPCR was used to measure miRNA expression both within EVs and in the surrounding matrix.

Overall, the experiments are well controlled. However, the manuscript would benefit from addressing the following critical points:

1.     A comparative analysis with other established EV isolation protocols should be presented, detailing the advantages and potential drawbacks of the method employed in this study.

2.     Given that Triton X-100 disrupts the bilipid membrane of EVs, the inclusion of a Triton X-100 treated sample as a control is essential for validating the isolation procedure.

3.     The quantification of individual miRNA expression across samples necessitates the use of both endogenous and exogenous controls for normalization. Without these controls, the raw miRNA counts lack substantive meaning.

Additional minor considerations include:

The citation format needs standardization, as the current manuscript incorrectly includes "[Internet]" following several journal names. The authors should revise and correct all citations accordingly.

Author Response

We appreciate the reviewer for the time and for the questions and comments to our work. We thank reviewer for good words to our manuscript. All the questions and remarks were carefully read by authors. As the result we have prepared the answers and performed some corrections into the text. We hope that all corrections are well done and we are sure that quality of the manuscript increased thanks to reviewer’s questions and remarks.

Answers to reviewer's questions

  1. A comparative analysis with other established EV isolation protocols should be presented, detailing the advantages and potential drawbacks of the method employed in this study.

We appreciate the reviewer for this important remark. Isolation of EVs from blood plasma or serum is still very challenging goal. Some claim that preparation of pure EVs remains an unachievable goal. All these despite the numerous methods of EVs isolation and their possible combinations. We prepared additional information about the importance of used method and brief description of available methods according to actual MISEV2018 (1)  and MIBLOOD-EV (2) recommendations. According reviewer's recommendations we added it to discussion section. We think that performed method of EVs isolation is of the MISEV2018 (1) and MIBLOOD-EV (2) recommendations and is adequate to the aim of the study.

Added to the discussion section

Blood-derived plasma and serum are the most commonly studied biofluids for extracellular vesicles research [51]. On the other hand blood is a complex fluid rich in soluble macromolecules (including lipoproteins), cells (intact and fragmented), and non-vesicular nucleic acids, that can strongly overlap with EVs in physical properties  and composition [51]. Thus, obtaining a pure EVs preparation, in particular from blood plasma, remains an unachieved goal[51]. Nevertheless, it is very challenging to separate EVs from blood plasma or serum to perform a study using one of the available isolation methods [7]. According to a recovery and specificity of achieving fraction, the current recommendations of the International Society for Extracellular Vesicles (MISEV2018) classifies all the methods for EV isolation into 4 groups [7]: 1. High recovery, low specificity (e.g. precipitation kits, low molecular weight cut-off centrifugal filters, very high speed ultracentrifugation without previous, lower-speed steps etc.); 2. Intermediate recovery, intermediate specificity (e.g. size exclusion chromatography, differential ultracentrifugation using intermediate time/ speed with or without wash, tangential flow filtration, and membrane-affinity columns etc.); 3. Low recovery, high specificity (immuno- or other affinity isolation including flow cytometry for large particles etc.); 4. High recovery, high specificity (may not be achievable as of the writing of the recommendations)  [7]. In the present study we performed the method of EVs isolation with initial consequential steps of centrifugation of fresh non-freeze plasma and following 0.8 µm filtering to eliminate platelets, fragmented platelets, erythrocytes as a major confounders of EV preparation [51, 52]. Besides we have performed high-speed centrifugation followed by washing procedures. Thus, used method may be qualified according MISEV2018 recommendations as intermediate recovery, intermediate specificity, allowing to recover mixed EVs along with some amount of free proteins and lipoproteins [7].

  1. Given that Triton X-100 disrupts the bilipid membrane of EVs, the inclusion of a Triton X-100 treated sample as a control is essential for validating the isolation procedure.

Lipid membrane disrupting control is known to be one of the mandatory to perform while investigating the extracellular vesicles. This statement is declared in all main recommendations of The International society for extracellular vesicles (1, 3, 4). Knowing that, we have performed these controls and demonstrated the results in the figures 6 and 8.    

  1. The quantification of individual miRNA expression across samples necessitates the use of both endogenous and exogenous controls for normalization. Without these controls, the raw miRNA counts lack substantive meaning.

We thank reviewer for pointing this interesting methodology aspect. Various methods for RT-qPCR miRNA data normalization, such as normalization to the geometric mean of all detected miRNAs, normalization to the endogenous control and the use of a synthetic spike-in have been described. However, there is still no consensus on the most appropriate normalization method: all methods have their pros and cons. We did not detect changes in expression between different groups of donors, but analyzed miRNAs levels from the same donors after a series of different treatments.. Actually, the spike-in that we add in Trizol at the RNA isolation step does not control the sampling step, but we tried to minimize the variability of this step through strict donor requirements and narrow timeframes for the material collection and the equipment used. Thus, we are sure that the chosen normalization strategy using spike-in is relevant for the presented study.

Additional minor considerations include:

The citation format needs standardization, as the current manuscript incorrectly includes "[Internet]" following several journal names. The authors should revise and correct all citations accordingly.

All bibliography citations were performed using Mendeley softwear and Vancouver citation style as it is recommended by the journal. Anyway we have checked all citation lists and corrected all links according to the appropriate standard.

Bibliography

  1. Théry C, Witwer KW, Aikawa E, Alcaraz MJ, Anderson JD, Andriantsitohaina R, et al. Minimal information for studies of extracellular vesicles 2018 (MISEV2018): a position statement of the International Society for Extracellular Vesicles and update of the MISEV2014 guidelines. J Extracell Vesicles. 2018;7(1).
  2. Lucien F, Gustafson D, Lenassi M, Li B, Teske JJ, Boilard E, et al. MIBlood-EV: Minimal information to enhance the quality and reproducibility of blood extracellular vesicle research. J Extracell Vesicles. 2023;12(12).
  3. Welsh JA, Van Der Pol E, Arkesteijn GJA, Bremer M, Brisson A, Coumans F, et al. MIFlowCyt-EV: a framework for standardized reporting of extracellular vesicle flow cytometry experiments. J Extracell Vesicles. 2020;9(1).
  4. Welsh JA, Van Der Pol E, Arkesteijn GJA, Bremer M, Brisson A, Coumans F, et al. MIFlowCyt-EV: a framework for standardized reporting of extracellular vesicle flow cytometry experiments. J Extracell Vesicles. 2020;9(1).

Reviewer 2 Report

Comments and Suggestions for Authors

                Extracellular vesicles (EVs) are an important mechanism of intercellular communication. MicroRNAs number among the large variety of bioactive molecules transported between cells by EVs. Currently, there is no established standard for EV isolation, making it difficult to compare experimental results from different research groups. This is because, depending on the preparation method, a variable amount of material externally coating the EVs will be co-precipitated. The authors of this work carefully documented this phenomenon and precisely quantified the miRNA which co-precipitates with EVs under different isolation conditions. They found that a modest but significant amount of miRNA co-precipitated with EVs which had not been treated with Proteinase K and RNase. This important finding is expected to have an impact on future research into EVs. Overall, this is an excellent paper which only needs some improvements to the English writing.

Comments on the Quality of English Language

The English is understandable, but there is a lot of incorrect grammar and/or awkward usage, particularly toward the end of the paper.

Author Response

We appreciate reviewer for the performed work. We are very pleased to receive such a good review on our manuscript. Besides, we are much more impressed because of reviewer’s comments about perspectives of the future research into EVs. It is very valuable for us because we are planning to continue the work.

We have carefully read the manuscript and performed some corrections related to English grammar. We hope that the quality of the paper improved.

Reviewer 3 Report

Comments and Suggestions for Authors

Petrova et al examined the copy number of several important miRNAs, (miR-30a, miR-223,  miR-let-7a, miR-let-7f, miR-451 and miR-486) in extracellular vesicles (EVs) of healthy donors. They developed the combined extraction method for differentiating components inside and adhered outside EVs and utilized WB, low voltage scanning electron microscopy, dynamic light scattering, nanoparticle tracking analysis and high-sensitivity flow cytometry approaches to illustrate and confirm the purity of EVs. The experiments are well designed and very elegant, while some details need to be touched on and shown. I have the following additional comments:

1.     In introduction section, some miRNAs with of pathological role in EVs should be highlighted.

2.     For miR-30a, miR-223, miR-let-7a, miR-let-7f, miR-451 and miR-486, the detailed introduction of these miRNAs should be involved, including the copy number, biogenesis, transportation mechanism etc.

3.     This manuscript detected the absolute number of these miRNAs in healthy donors. What about the patients, especially in the miRNAs associated diseases? Is it similar with healthy donor?

4.     The authors calculated the molecular ratio of miRNA inside and adhered EVs. It would be interesting to calculate the molecular ratio of EV and inside cells.

Author Response

We appreciate the reviewer for the time and for the questions and comments to our work. We thank reviewer for good words to our manuscript. All the questions and remarks were carefully read by authors. As the result we have prepared the answers and performed some corrections into the text. We hope that all corrections are well done and we are sure that quality of the manuscript increased thanks to reviewer’s questions and remarks. 

Answers to the questions

  1. In introduction section, some miRNAs with of pathological role in EVs should be highlighted.

We have added information about pathological role of miRNAs associated with EVs in the Introduction section.

Addition to the text.

Besides, some miRNAs demonstrate their pathological role in EVs. For instance, miRNA 105 secreted in breast cancer cell exosomes disrupts epithelial monolayer tight junctions, promoting metastasis [21]. EVs released by the lung cancer-activated mast cells contain significantly increased amounts of protumorogenic miRNA 100 and miRNA 125b, which regulate p53 signaling, cancer pathways and pathways related to apoptosis and cell cycle [22].

  1. For miR-30a, miR-223, miR-let-7a, miR-let-7f, miR-451 and miR-486, the detailed introduction of these miRNAs should be involved, including the copy number, biogenesis, transportation mechanism etc.

Taking into account the significance of the known information about investigated miRNAs we have added literature data to the Discussion section.

Addition to the text

In current study we analyzed the absolute number of miRNA (hsa-30a, hsa-223, hsa-let-7a, hsa-let-7f, , hsa-451, hsa-486, ) inside and attached to the outside of plasma EVs. MiRNA-30a was identified by Lagos-Quintana and colleagues and is widely distributed in humans, accounting for 4.1% of the total number of miRNAs expressed [68]. It originates from an intronic transcriptional unit and is involved in a wide range of biological processes, including cell differentiation and development, and plays a dual role as an oncogene or oncosuppressor in various cancers [42,69]. miRNA 223 is transcribed from an independent promoter and is highly expressed in the hematopoietic system [39,70,71]. MiRNA 223 expression changes considerably during cellular differentiation. It increases during osteoclast, megakaryocyte and eosinophil differentiation and decreases during erythrocyte and macrophage differentiation [72]. The highly conserved let-7 miRNA family includes, among others, let-7a and let-7f miRNAs, which are clustered together. The let-7 family of miRNAs is expressed during embryogenesis and brain development and is a tumor suppressor [73]. MiRNA let-7a and let-7f were identified by Lagos-Quintana and colleagues and have canonical biogenesis pathway [68]. MiRNA let-7a has 3 genes in human genome and is highly expressed in skin, submandibular gland. miRNA let-7f has 2 genes in genome and is highly expressed in sclera and submandibular gland [74]. miRNA 451 has 1 gene in genome and is highly expressed in red blood cells, bones [70]. miRNA 451 has noncanonical Drosha/DGCR8-dependent/Dicer independent biogenesis [75,76]. MiRNA 486 is transcribed from an intron within the ANK1 locus [70,77]. It has high expression in skeletal and cardiac muscle, erythroid cells and circulates at high levels in plasma [41]. MiRNA 486 is enriched within small extracellular vesicles and has atypical biogenesis [78,79].

  1. This manuscript detected the absolute number of these miRNAs in healthy donors. What about the patients, especially in the miRNAs associated diseases? Is it similar with healthy donor?

There are limited papers investigating the ratio of miRNAs per EVs, especially in pathological conditions. In most cases authors demonstrate differential expression of targeted miRNAs in pathology vs. normal conditions, suggesting their impact in pathogenesis and proposing potential diagnostic value. Meanwhile we propose that not only the level of miRNAs but also their topography according extracellular vesicles may influence on pathogenesis of various diseases. Thus, we started from the investigation of topography distribution of miRNAs in physiological conditions planning to continue investigations of those in pathological conditions.

We have added existing literature results in discussion section

There are limited papers investigating the ratio of miRNAs per EVs, especially in pathological conditions. Thus, Chevillet and colleagues showed that even abundant miRNAs are present in amounts much less than one copy per exosome even in physiological conditions [62]. Besides, the numbers of copies of miRNA per exosome for  miRNA-126 and miRNA-223 in patients with prostate cancer were higher than those in healthy donors [62]. Anyway we can propose that the number of miRNAs transported by EVs could be of significance in pathogenesis of various diseases.

  1. The authors calculated the molecular ratio of miRNA inside and adhered EVs. It would be interesting to calculate the molecular ratio of EV and inside cells.

We suggest that this will be an interesting challenge to calculate the ratio of miRNAs per single cell inside and outside the membrane, especially taking into account difficult topography of cells and their membranes. But for now we are focusing on investigating extracellular vesicles in physiological and pathological conditions. Anyway we shall keep in mind this opportunity.  

Round 2

Reviewer 1 Report

Comments and Suggestions for Authors

The authors have addressed all of my concerns satisfactorily

Reviewer 3 Report

Comments and Suggestions for Authors

Thanks for the additional information.